# Diffusion Process with Implicit Latents via Energy Models

## Abstract

We present a generative model based on an ordered sequence of latent variables for intermediate distributions between a given source and a desired target distribution. We construct the probabilistic transitions among the latent variables using energy models that are in the form of classifiers. In our work, the intermediate transitional distributions are implicitly defined by the energy models during training, where the statistical properties of the data distribution are naturally taken into account. This is in contrast to denoising diffusion probabilistic models (DDPMs) where they are explicitly defined by the predefined scheduling of a sequential noise degradation process. Over the course of training, our model is designed to optimally determine the intermediate distributions by Langevin dynamics driven by the energy model. In contrast, energy-based models (EBMs) typically require an additional generator since the intermediate distributions are not explicitly defined in the training procedure. We demonstrate the effectiveness and efficiency of the proposed algorithm in the context of image generation, achieving high fidelity results with less inference steps on a variety of datasets.

## 1 Introduction

Learning generative models for a data distribution is considered a significant problem in machine learning and its different applications, such as computer vision and language models. A variety of algorithms have been developed for image generation, including Variational Autoencoders (VAEs) (Kingma (2013)), Generative Adversarial Networks (GANs) (Goodfellow et al. (2014); Karras et al. (2020)), Diffusion-based Models (Ho et al. (2020); Dhariwal & Nichol (2021)) and Energy-based Models (Du & Mordatch (2019); LeCun et al. (2006)).

One of the most widely used algorithms are Denoising Diffusion Probabilistic Models (DDPMs) (Ho et al. (2020)). DDPM defines a forward process by adding noise to the data and trains models for its reverse process, leading to sequential generative steps. One of the drawbacks of DDPM, and its variants, is the computational inefficiency due to the necessity for a large number of sampling steps. This is because intermediate distributions are defined by the scheduled noise process, which requires a sufficiently small transitional distribution gap to account for the variability of samples to generate (Xiao et al. (2021)). Albeit, there have been attempts to accelerate sampling based on non-Markovian diffusion processes (Kong & Ping (2021); Song et al. (2020). However, it is desirable to consider the variability of the data in the determination of scheduling for transitional distributions and the local measure of data information in the spatial domain, leading to our adaptive determination of latent distributions via energy models.

Another branch of generative learning algorithms are energy-based models (EBMs) (Du & Mordatch (2019); Gao et al. (2020b)), that aim to learn explicit probability distribution from the data in terms of energy functions associated with model parameters. The energy function is designed to assign energy values to data samples based on contrastive divergence learning, in which the optimization is often computationally intensive and unstable since it involves a sampling process with Markov chain Monte Carlo (MCMC). Because of this, it is generally required to employ a generator, that is guided by the energy function to map an element in the latent space to a sample closer the data space (Xiao et al. (2020); Xie et al. (2022); Cui & Han (2023)). However, the latent variable for each transitional distribution is not specified by the energy function.

In this work, we develop a generative model based on a sequence of energy functions learned by a time-conditioned classifier, designed to identify the intermediate latent distributions. These distributions are adaptively determined by the energy functions, in consideration of statistical properties of the data distribution. The training of the energy functions follows the stochastic gradient Langevin process (Welling & Teh (2011); Nijkamp et al. (2019)) and constructs a sequence of intermediate distributions without specifying their associated statistical properties, such as the mean and variance in the case of a normal distribution. Thus, the proposed algorithm does not require a predefined schedule of the diffusion process, and furthermore, the transitional step in the diffusion process is determined by the energy function in an adaptive way by considering the distributional discrepancy between current sampling and the data. The training procedure of our algorithm leads the energy model to define implicit latent variables in such a way that the distributional transition gap is optimally arranged by the Langevin steps. In our algorithm, one temporal sweep over time in the training process is identical to the sampling process, implying simplicity and efficiency. In the application of contrastive learning, we employ a regularization term based on the gradient penalty for a Lipschitz constraint to achieve more stable optimization and better generalization (Gulrajani et al. (2017); Petzka et al. (2017)).

We present quantitative and qualitative comparisons to both diffusion-based models and energy-based models in image generation tasks. Our algorithm can effectively learn a variety of data distributions and generate competitive samples in a significantly reduced number of inference steps, compared to conventional methods.

## 2 RELATED WORK

**Diffusion models**    Diffusion probabilistic models are a family of generative models, introduced by Sohl-Dickstein et al. (2015), that learn a data distribution by reversing an iterative noise degradation process. Thanks to a number of advancements since then (Ho et al. (2020); Nichol & Dhariwal (2021); Dhariwal & Nichol (2021)), denosing diffusion probabilistic models (DDPM) achieved incredibly high-quality results in a variety of image synthesis tasks. However, to generate images with these models takes a notoriously large number of sampling iterations, and there is a lot of published work on the topic of reducing diffusion model inference times (Xiao et al. (2021); Wang et al. (2022); San-Roman et al. (2021); Kong et al. (2020)). Notably, Song et al. (2020) propose denoising diffusion implicit models (DDIM), which employ a non-Markovian degradation process that lends to more efficient sampling, while still preserving the original training objective of DDPM.

Our method differs from previous approaches because we do not apply any degradation to the real data distribution, which means that all the intermediate latent distributions are learned implicitly by the energy model. This is advantageous, as degradation, such as Gaussian noise, may fail to capture the full multi-modal data distribution at large step sizes Xiao et al. (2021). Additionally, both DDIM (Song et al. (2020)) and DDPM Ho et al. (2020) define intermediate latent distributions as a linear interpolation that is equally applied to all image pixels. This can be inefficient in representing the true data distribution, as there are typically image regions with greater semantic significance than others. Our method avoids these issues because the intermediate distributions are defined by the, more flexible, energy model.

**Energy-Based Models.**    In the field of machine learning, early studies on EBMs have demonstrated their promising generative capabilities (LeCun et al. (2006)). Tieleman (2008) introduced persistent contrastive divergence (PCD) that is still commonly used today. Du & Mordatch (2019) have shown that EBMs can be successfully scaled to modern deep neural networks, and Nijkamp et al. (2019) proved that a finite number of Langevin dynamics iterations is enough to generate high quality samples from the EBM. However, the difficulty to approximate Boltzmann's distribution using MCMC sampling, remains as the main hindering factor when compared to other generative approaches. To overcome this, Yang & Ji (2021) and Yang et al. (2023) begin sampling from a latent distribution that is closer to the target than why noise by informed initialization. Similarly, Zhao et al. (2016), Xiao et al. (2020), Cui & Han (2023) and Han et al. (2019) use a generator model to initialize the sampling process, thus skipping the more difficult MCMC steps performed on the noisiest data. Our method aims to better guide sampling by learning a sequence of energy functions at intermediate latent distributions. This alleviates the need for additional generators, as our model is able to learn the appropriate energy landscapes even at the early steps of the sampling process.

Gao et al. (2020b) also combine EBMs with diffusion models by training a sequence of EBMs. They learn the recovery likelihoods that are defined by the intermediate latent distributions of a predefined noise diffusion process. Our method differs in that we do not specify any noise diffusion process, and instead allow the energy model to learn the latent distributions implicitly. Additionally. instead of training for the recovery likelihood we employ the contrastive divergence training object, that is used more commonly in EBMs.

## 3 PRELIMINARIES

We provide background on denoising diffusion probabilistic models (DDPMs) and Energy-Based models (EBMs), which are closely related to the construction of our algorithm.

### 3.1 DIFFUSION-BASED MODELS

Let $\{x_t \mid t = 1, 2, \cdots, T\}$ be a set of sequential latent variables in the sample space $\chi$ associated with a temporal variable $t$. The DDPMs are latent variable models of the form:

$$p_\theta(x_0) = \int p_\theta(x_{0:T}) dx_{1:T}, \quad p_\theta(x_{0:T}) = p(x_T) \prod_{t=1}^{T} p_\theta(x_{t-1} \mid x_t), \quad p(x_T) = \mathcal{N}(x_T; 0, I), \quad (1)$$

where $x_0 \sim q(x_0)$ and the joint distribution $p_\theta(x_{0:T})$ is defined as a Markov chain of the backward process $p_\theta(x_{t-1} \mid x_t)$ with an initial distribution $p(x_T)$. The backward process $p_\theta(x_{t-1} \mid x_t)$ is defined by a Gaussian transition with its associated learnable parameters $\mu_\theta$ and $\Sigma_\theta$ as follows:

$$p_\theta(x_{t-1} \mid x_t) = \mathcal{N}(x_{t-1}; \mu_\theta(x_t, t), \Sigma_\theta(x_t, t)). \quad (2)$$

The approximate posterior is given by a conditional joint distribution $q(x_{1:T} \mid x_0)$ defined by a Markov chain of the forward process $q(x_t \mid x_{t-1})$ that is designed to add Gaussian noise with the scheduled variance $\beta_t \in (0, 1)$ as follows:

$$q(x_{1:T} \mid x_0) = \prod_{t=1}^{T} q(x_t \mid x_{t-1}), \quad q(x_t \mid x_{t-1}) = \mathcal{N}(x_t; \sqrt{1 - \beta_t} x_{t-1}, \beta_t I), \quad (3)$$

where $\beta_t$ is scheduled to increase over time $t$ so that the latent $x_T$ in the forward process becomes to follow a Gaussian distribution of the initial distribution $p(x_T)$ in the backward process. The objective of training the latent variable model $p_\theta(x_{0:T})$ for a sequential generative process from $x_T$ to $x_0$ is to maximize the log-likelihood $\log p_\theta(x_0)$ leading to minimizing Kullback-Leibler divergence between forward and backward processes as follows:

$$D_{KL}(q(x_{t-1} \mid x_t, x_0) \,\|\, p_\theta(x_{t-1} \mid x_t)). \quad (4)$$

### 3.2 ENERGY-BASED MODELS

Energy-based models are designed to represent an energy function $f_\theta(x) \in R$ that outputs high values if $x$ belongs to a given data distribution, and low values if it does not. In the The probability density function $p_\theta(x)$ for a EBM is defined via Boltzmann's distribution as given by:

$$p_\theta(x) = \frac{\exp(-f_\theta(x)))}{Z(\theta)}, \quad (5)$$

where $Z_\theta(x) = \int \exp(f_\theta(x)) \, \mathrm{d}x$ is the partition function used for normalization.

Since the computation of the partition function $Z_\theta(x)$ is intractable, direct sampling from $p_\theta(x)$ is often infeasible, which results in a significant computational challenge of training EBMs. There have been a number of sampling approaches such as Markov chain Monte Carlo or Gibbs sampling in order to approximate the distribution density. One of the most widely used algorithms is Stochastic Gradient Langevin Dynamics (SGLD) leading to the following update:

$$x_{t+1} = x_t - \frac{\eta}{2} \frac{\partial}{\partial x_t} f_\theta(x_t) + \sqrt{\eta} \, \epsilon_t, \quad \epsilon_t \sim \mathcal{N}(0, I), \quad (6)$$

where $\eta$ denotes the step size and the variance of the Gaussian noise perturbation, and $\epsilon_t$ follows a standard normal distribution.

The training of the associated model parameters $\theta$ is achieved in the framework of maximum a posteriori (MAP) leading to the derivative of the log-likelihood for a target real distribution $q$ as defined by:

$$\frac{\partial}{\partial \theta} \log p_\theta(x) = \mathbb{E}_{x \sim q} \left[ \frac{\partial f_\theta(x)}{\partial \theta} \right] - \mathbb{E}_{x \sim p_\theta} \left[ \frac{\partial f_\theta(x)}{\partial \theta} \right], \tag{7}$$

which is the derivative of a contrastive divergence loss between the target $q$ and an estimate $p_\theta$.

## 4 METHOD

In our proposed algorithm, we construct a sequence of intermediate distributions represented by latent variables from a given source distribution to the target following the stochastic Langevin process similar to the algorithms in Gao et al. (2020b); Du et al. (2024). However our, we construct them in an implicit way, driven by the energy functions, without an explicit scheduling of the distributions. In the formulation of the objective function we consider a regularization term that utilizes gradient penalty (Gulrajani et al. (2017); Petzka et al. (2017)), thus ensuring the numerical stability of the optimization.

### 4.1 GENERATIVE PROCESS

Let $f_\theta \colon \mathcal{X} \times [1, T] \mapsto \mathbb{R}$ be a real-valued energy function where $T$ is a given number of time steps and $\theta$ denotes a set of model parameters. The energy function $f_\theta(x, t)$ takes a pair consisting of a latent variable $x_t \in \mathcal{X}$ and its associated time step $t \in [1, T]$ and aims to construct a sequence of distributions from a known distribution $p(x_T) = \mathcal{N}(x_T; 0, I)$ to an approximate $p_\theta(x_0) \approx q(x_0)$. The probability density function for $x_t$ at time step $t$ is defined by the Boltzmann distribution as follows:

$$p_\theta(x_t) = \frac{1}{Z_{\theta,t}} \exp\left(-f_\theta(x_t, t)\right), \quad Z_{\theta,t} = \int \exp\left(-f_\theta(x_t, t)\right) \, \mathrm{d}x_t, \tag{8}$$

where $Z_{\theta,t}$ is the partition function that is obtained by integrating over the intermediate distribution of latent variable $x_t$. It is computationally infeasible to evaluate the partition function and we approximate distribution $p_\theta$ using MCMC technique leading to the following Langevin step that defines the backward process of the algorithm:

$$p_\theta(x_{t-1}|x_t) = \mathcal{N}(x_{t-1}; \tilde{x}_t, \eta I), \quad \tilde{x}_t = x_t - \frac{\eta}{2} \nabla_x f_\theta(x_t, t), \tag{9}$$

where $\eta$ denotes the variance of the Gaussian noise in the Langevin process and the learning rate of the gradient descent. In contrast to the algorithms in DDPMs where both the forward and backward processes are defined as given in equation 3 and equation 2, respectively, our generative process is developed based on the backward process in which intermediate distributions for latent variables are implicitly specified as the training of the energy model $f_\theta(x_t, t)$ proceed with the assumption that the forward process is constant as defined by:

$$q(x_t|x_{t-1}) = \mathcal{N}(x_t; x_{t-1}, 0), \tag{10}$$

where $q(x_0)$ represents the observed data distribution and we assume that the desirable distribution $q(x_t)$ at each time step $t$ is the same as the observed data distribution $q(x_t) \approx q(x_0)$ for any $t$. Thus, the estimation of intermediate distribution $p_\theta(x_t)$ represented by latent variable $x_t$ is performed in such a way that an estimate $p_\theta(x_t)$ is pushed toward the desirable distribution $q(x_0)$ for any $t$.

### 4.2 OBJECTIVE FUNCTION

The training of energy model the $f_\theta(x_t, t)$ is performed by maximizing the log-likelihood $\log p_\theta(x_0)$ for an observed distribution $x_0 \sim q(x_0)$ in the form of marginal probability over the latent variables

as defined by:

$$\log p_\theta(x_0) = \log \int p_\theta(x_0, x_1, \cdots, x_T) \, dx_1 \, dx_2 \cdots dx_T = \log \int \frac{q(x_{1:T}|x_0)}{q(x_{1:T}|x_0)} p_\theta(x_{0:T}) \, dx_{1:T}$$

$$= \log \int q(x_{1:T}|x_0) p(x_T) \frac{p_\theta(x_{T-1}|x_T) \cdots p_\theta(x_0|x_1)}{q(x_T|x_{T-1}) \cdots q(x_1|x_0)} \, dx_{1:T}$$

$$= \log \int q(x_{1:T}|x_0) p(x_T) \frac{p_\theta(x_{T-1}|x_T) \cdots p_\theta(x_0|x_1)}{q(x_T) \cdots q(x_1)} \, dx_{1:T}, \tag{11}$$

where we assume that $x_1, \cdots, x_T$ are independent and identically distributed random variables with a constant distribution $q(x_0) = q(x_1) = \cdots = q(x_T)$ as defined in equation 10. The objective of optimization is to minimize the evidence lower bound of the negative log-likelihood given by:

$$\mathbb{E}_q \left[ -\log p(x_T) \frac{p(x_{T-1}|x_T) \cdots p_\theta(x_0|x_1)}{q(x_T) \cdots q(x_1)} \right]$$

$$= \mathbb{E}_q \left[ -\log p_\theta(x_0|x_1) - \log \frac{p(x_T)}{q(x_T)} - \sum_{t=1}^{T-1} \log \frac{p_\theta(x_t|x_{t+1})}{q(x_t)} \right] \tag{12}$$

$$= \mathbb{E}_q \left[ -\log p_\theta(x_0|x_1) \right] + D_{KL}(q(x_T) \,\|\, p(x_T)) + \sum_{t=1}^{T-1} D_{KL}(q(x_t) \,\|\, p_\theta(x_t|x_{t+1})),$$

where $D_{KL}(q(x_T) \,\|\, p(x_T))$ is constant with respect to $\theta$ and we have:

$$\mathbb{E}_q \left[ -\log p_\theta(x_0|x_1) \right] = -q(x_1|x_0) \log p_\theta(x_0|x_1) = -q(x_1|x_0) \log p_\theta(x_0|x_1) \frac{q(x_0)}{q(x_0)}$$

$$= -q(x_1|x_0) \left( \log \frac{p_\theta(x_0|x_1)}{q(x_0)} + \log q(x_0) \right) = -q(x_0) \left( \log \frac{p_\theta(x_0|x_1)}{q(x_0)} + \log q(x_0) \right) \tag{13}$$

$$= D_{KL}(q(x_0) \,\|\, p_\theta(x_0|x_1)) + H(q(x_0)),$$

where the entropy $H(q(x_0))$ of the observed distribution $q(x_0)$ is constant with respect to $\theta$, thus we have the following objective function:

$$\mathcal{L}(\theta) = \sum_{t=0}^{T-1} D_{KL}(q(x_t) \,\|\, p_\theta(x_t|x_{t+1})). \tag{14}$$

The objective function computes the distributional discrepancy between the desirable distribution $q(x_t)$ and its corresponding estimate $p_\theta(x_t|x_{t+1})$ conditioned by its previous state $x_{t+1}$ at any $t$.

### 4.3 TRAINING

The training procedure consists of two alternating phases, one of which is aimed to optimize energy function $f_\theta(x_t, t)$ with respect to its associate model parameters $\theta$ and the other is to improve the estimation of density $p_\theta(x_t)$. The optimization for the energy function is performed by taking the gradient of the objective function $\mathcal{L}$ in equation 14 with respect to $\theta$ as defined by:

$$\theta^{\tau+1} = \theta^\tau - \xi^\tau \nabla_\theta \mathcal{L}(\theta^\tau), \tag{15}$$

where $\tau$ denotes the index of the gradient descent steps, $\xi^\tau$ is the learning rate and the computation of the gradient $\nabla_\theta \mathcal{L}(\theta^\tau)$ at $\theta^\tau$ reads:

$$\nabla_\theta \mathcal{L}(\theta^\tau) = \sum_{t=0}^{T-1} \nabla_\theta \ell_t(\theta^\tau), \quad \nabla_\theta \ell_t(\theta^\tau) = \mathbb{E}_{x_t \sim q} \left[ \nabla_\theta f_\theta(x_t, t) \right] - \mathbb{E}_{x_t \sim p_\theta} \left[ \nabla_\theta f_\theta(x_t, t) \right], \tag{16}$$

which leads to the gradient of the contrastive divergence loss between $q$ and $p_\theta$ at $t$. The evaluation of the first term in equation 16 involves sampling $x_t \sim p_\theta(x_t)$ from the energy model $f_\theta$ using Langevin dynamics by the gradient descent as follows:

$$x_{t-1} = x_t - \frac{\eta}{2} \nabla_x f_\theta(x_t, t) + \sqrt{\eta} \, \epsilon_t, \tag{17}$$

Figure 1: The full sampling process for generating images from the CelebA dataset of size $64 \times 64$.

where $\eta$ is the step size of the Langevin process and $\epsilon_t \sim \mathcal{N}(0, I)$. We assume that the maximum number $T$ of latent variables $\{x_t \,|\, t = 1, 2, \cdots, T\}$ is given to the training in which the cyclic constraint $t := T$ when $t = 0$ is applied in the sequential update over decreasing order of time steps $t := t - 1$. The training procedure repeats stochastic updates of estimates for $x_t$ with a refresh condition $p(x_T) = \mathcal{N}(x_T \,;\, 0, I)$. We also set the number of Langevin steps defined in equation 17. Consequently, a decreasing annealing scheme has to be applied to the variance of the noise term $\epsilon_t$, as described by Nijkamp et al. (2019).

## 4.4 REPLAY BUFFER

When training energy models many works employ Persistent Contrastive Divergence (PCD) (Tieleman (2008)). This allows for refining previously synthesized samples during the training process, without the need to repeat the costly MCMC sampling process in full (Du et al. (2020a); Yang & Ji (2021)). We adopt a similar approach, but with the goal of training for the intermediate distributions $x_t$.

We define a sample buffer $\mathcal{B}$ that consists of a pair of generated samples $x$ and their associated time steps $t$. In the beginning of training, the data in the initial buffer is assigned with random samples $x \sim \mathcal{N}(x \,;\, 0, I)$ and their associated time steps are also randomly assigned as $t \sim U(1, T)$ where $U(1, T)$ denotes a uniform distribution of integers between 1 and $T$. At each iteration of the Langevin process in equation 17, a batch of data $x$ is randomly taken from the sample buffer $\mathcal{B}$ and their time steps $t := t - 1$ are decreased by 1 with a cyclic constraint $t := T$ when $t = 0$, assigning $x \sim \mathcal{N}(x \,;\, 0, I)$ for re-initialization.

## 4.5 GRADIENT PENALTY

The contrastive divergence (CD) loss in equation 16 is known to be unstable, and is usually paired with regularization techniques that aim to impose the 1-Lipschitz constraint on model parameters. Gulrajani et al. (2017) introduced gradient penalty as a soft regularization:

$$\mathbb{E}_{\hat{x} \sim \gamma} \left[ (||\nabla_{\hat{x}} f_\theta(\hat{x})||_2 - 1)^2 \right], \tag{18}$$

where $\gamma$ is the distribution of $\hat{x} = \alpha x^- + (1 - \alpha)x^+$, where $x^- \sim p_\theta$, $x^+ \sim q$ and $\alpha \in U(0, 1)$.

In addition to regularizing the training process, gradient penalty also restricts the gradients when sampling through equation 17. This leads to better stability, especially in earlier sampling steps. We observed that another popular regularization technique, spectral normalization (Miyato et al. (2018)), doesn't prevent large gradients in the sampling process. Gradient penalty also balances the loss magnitudes of our model at different time steps by penalizing larger losses more harshly.

Equation 18 restricts the gradient to be 1 across all time steps. This is undesirable as it prevents convergence of the algorithm. Thus, we apply a modified gradient penalty, named WGAN-LP, described by Petzka et al. (2017) as:

$$\mathbb{E}_{\hat{x} \sim \gamma} \left[ (\max\{0, ||\nabla_{\hat{x}} f_\theta(\hat{x})||_2 - 1\})^2 \right], \tag{19}$$

which enforces the gradient to be less than or equal to 1. The full training process is summarized in Algorithm 1 where we omit the time sample buffer and batched data for ease of presentation. Finally, a visual illustration of the sampling process is presented in Fig. 1

---

**Algorithm 1** Training algorithm

---

**Input:** data dist. $q$, sampling step size $\eta$, total time steps $T$, number of SGLD steps $K$, noise variance $\sigma$ and gradient penalty weight $\lambda$.
**Initialize:**
    $t \sim U(1, T)$
    $x^- \sim \mathcal{N}(0, I)$
**while** not converged **do**
    **for** $k \leftarrow 1$ to $K$ **do**
        $x^- \leftarrow x^- - \frac{\eta}{2} \nabla_{x^-} f_\theta(x^-, t) + \mathcal{N}(0, \sigma^2 I)$          ▷ Fake sample update
    **end for**
    Sample $x^+ \sim q$ and $\alpha \sim U(0, 1)$
    $\hat{x} \leftarrow \alpha x^- + (1 - \alpha)x^+$
    $\nabla\theta \leftarrow \nabla_\theta (f_\theta(x^-, t) - f_\theta(x^+, t) + \lambda(\max\{0, \|\nabla_{\hat{x}} f_\theta(\hat{x})\|_2 - 1\})^2)$
    Update $\theta$ according to $\nabla\theta$ and Adam optimizer.
    $t \leftarrow t - 1$
    **if** $t = 0$ **then**          ▷ Refresh samples that reached the final sampling step.
        $t \leftarrow T$
        $x^- \sim \mathcal{N}(0, I)$
    **end if**
**end while**

---

Table 1: Comparisons of our method with previous generative models on CIFAR-10.

| Model | FID↓ |
|---|---|
| **Generative adversarial networks** | |
| DCGAN Radford et al. (2015) | 37.11 |
| WGAN + GP Gulrajani et al. (2017) | 36.4 |
| SNGAN Miyato et al. (2018) | 21.7 |
| StyleGAN2-ADA Karras et al. (2020) | 3.26 |
| **Score-based models** | |
| NCSN Song & Ermon (2019) | 25.32 |
| NCSN-v2 Song & Ermon (2020) | 10.87 |
| DDPM Ho et al. (2020) | 3.17 |
| **Energy-based models** | |
| Short-run EBM Nijkamp et al. (2019) | 44.50 |
| IGEBM (ensemble) Du & Mordatch (2019) | 38.2 |
| Flow Contrastive EEBM Gao et al. (2020a) | 37.3 |
| JEM++ Yang & Ji (2021) | 37.1 |
| Divergence Triangle Han et al. (2019) | 30.10 |
| EBM-BB Geng et al. (2021) | 28.63 |
| ImprovedCD EBM Du et al. (2020b) | 25.1 |
| GEBM Arbel et al. (2020) | 23.02 |
| Ours-Small | 18.05±0.09 |
| Ours | 17.03±0.08 |

Table 2: Comparisons with other methods on CelebA64[2]

| Model | FID↓ |
|---|---|
| DCGAN Radford et al. (2015) | 38.39 |
| COCO-GAN Lin et al. (2019) | 4.0 |
| NCSN Song & Ermon (2019) | 25.30 |
| NCSN-v2 Song & Ermon (2020) | 10.23 |
| Divergence Triangle Han et al. (2019) | 24.7 |
| FC-EBM Gao et al. (2020a) | 12.21 |
| CF-EBM Zhao et al. (2020) | 10.80 |
| Ours | 8.05±0.04 |

Table 3: Comparison of our models to the baseline IGEBM Du & Mordatch (2019) in number of parameters, training GPU hours and sampling time (for 50k samples of 32x32 images)

| Model | Parameters | Training | Sampling |
|---|---|---|---|
| IGEBM | 5M | 48h | 3h |
| Ours-Resnet | 6M | 48h | 0.24h |
| Ours | 9M | 96h | 0.37h |

## 5 EXPERIMENTS

**Implementation details.** We implement our model using the encoder part of the time-conditioned Unet architecture used in DDPM (Ho et al. (2020)). In order to avoid gradient artifacts, all down-sampling convolution layers were replaced with sub-pixel pooling operations. We also found it crucial to use Sigmod activations and to avoid using attention layers altogether, this is likely because our model is more reliant on smooth gradients during sampling. Unlike in previous energy based models (Nichol & Dhariwal (2021); Gao et al. (2020b)) we found normalization to be beneficial in training, and so we employ Layer Normalization (Lei Ba et al. (2016)).

**Hyperparameters.** We utilize WGAN-LP with a fixed weight of $\lambda$=200 in all experiments. Petzka et al. (2017) highlighted that such a high weight does not significantly hinder performance, and we found it helpful for achieving consistent sampling gradients across different configurations.

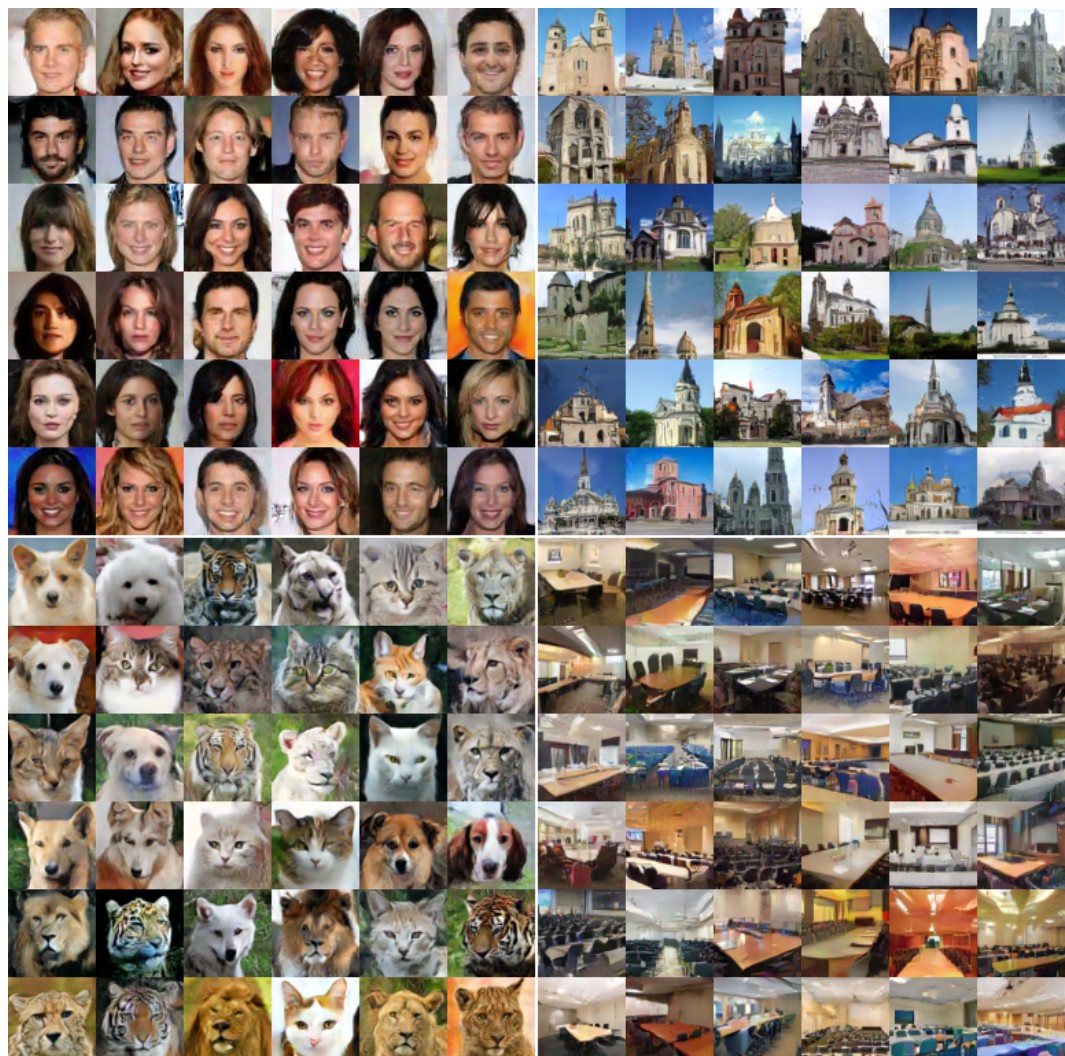

Figure 2: 64x64 resolution samples generated by our model from CelebaHQ (top-left), LSUN Churches (top-right), AFHQV2 (bottom-left) and LSUN Conference Room (bottom-right).

Additionally, all experiments use the AdamW optimizer (Loshchilov & Hutter (2017)) with $\beta_1 = 0.9$ and $\beta_2 = 0.999$, cosine annealing schedule with warm restarts (Loshchilov & Hutter (2016)) at a starting learning rate of $2 \times 10^{-4}$ and EMA with a rate of 0.999. We set a batch size of 256 on CIFAR-10 and 128 on all other datasets. The sample buffer described in Section 4.4 is always initialized to contain 10000 samples. The sampling step size $\eta$ is kept constant over all time steps, and we adjust it according to the total number of steps and noise variance in each experiment.

In Table 4 and Table 5 we ablate different combinations of the number of time steps $T$ and iterations of Langevin dynamics at each step $K$. For evaluation we present both the FID (Heusel et al. (2017)) and Inception Score (Salimans et al. (2016)) metrics. Our model is optimal when the total number of sampling steps is around 60 ($T \times K = 60$), and performance drops significantly as either parameter is increased. This finding is consistent with many previous EBMs (Du & Mordatch (2019); Gao et al. (2020b)), which may be due to the inaccuracies of MCMC sampling that accumulate over larger numbers of iterations. In Table 6 we present results for different initial values of the noise variance $\sigma$, that is always linearly annealed to 0 during sampling. Our model seems to benefit more from slightly higher values of $\sigma$ than previous, comparable, works like Nijkamp et al. (2019).

Figure 3: Interpolation results between the leftmost and rightmost generated images on CelebA.

Table 4: Ablation over the number of time steps $T$ for fixed $K = 3$ and $\sigma = 0.3$.

Table 5: Ablation over the number of Langevin steps $K$ for fixed $T = 20$ and $\sigma = 0.3$.

Table 6: Ablation over the noise variance sigma $\sigma$ for $T = 20$, $K = 3$.

| T | FID↓ | Inception ↑ |
|---|---|---|
| 6 | 22.18 | $6.86 \pm 0.06$ |
| 10 | 19.81 | $7.04 \pm 0.11$ |
| 20 | **18.05** | $\mathbf{7.32 \pm 0.11}$ |
| 30 | 24.21 | $7.06 \pm 0.08$ |
| 40 | 32.75 | $6.93 \pm 0.08$ |

| K | FID ↓ | Inception ↑ |
|---|---|---|
| 1 | 38.57 | $6.55 \pm 0.09$ |
| 3 | **18.05** | $\mathbf{7.32 \pm 0.11}$ |
| 5 | 21.86 | $6.91 \pm 0.08$ |
| 10 | 28.74 | $6.44 \pm 0.09$ |
| 30 | 31.75 | $6.53 \pm 0.11$ |

| $\sigma$ | FID ↓ | Inception ↑ |
|---|---|---|
| 0.005 | 29.45 | $6.37 \pm 0.09$ |
| 0.05 | 24.8 | $6.88 \pm 0.09$ |
| 0.1 | 22.09 | $6.85 \pm 0.08$ |
| 0.3 | **18.05** | $\mathbf{7.32 \pm 0.11}$ |
| 0.5 | 38.77 | $6.59 \pm 0.07$ |

## 5.1 IMAGE GENERATION

In Table 1 we compare our best FID score on the Cifar-10 Krizhevsky et al. (2009) dataset, where our model achieves an average FID of 17.03. Additionally, we resent results for a lighter Resnet-based implementation, marked as "Ours-small", for better comparison with previous EBMs. This Resnet network achieved an FID score that is $52.75\%$ better than its equivalent in IGEBM Du & Mordatch (2019). In table 2 we compare results on the CelebA (Liu et al. (2015)) dataset, where our model scores an average FID of 8.05. When training on CelebA, we follow the preprocessing approach of Zhao et al. (2020) and perform a center crop of $140\times140$ pixels before resizing each image to a $64\times64$ resolution. We calculate all metrics on a sample size of 50k unconditionally generated images. In Table 3 we compare the computational overhead of our models to the original IGEBM approach, showcasing that our method is able to learn a computationally more efficient sampling process than a traditional energy-based model.

Qualitative results for samples generated in a $64\times64$ resolution are shown in Figure 2. Our model demonstrates capabilities of synthesizing images from a variety of datasets; including CelebAHQ (Karras et al. (2017)), LSUN conference rooms , LSUN churches (Yu et al. (2015)) and AFHQV2 (Choi et al. (2020)) . Figure 1 displays the full sampling process of our model, highlighting the implicitly learned latent distributions. Our model is also capable of smooth interpolations between generated images, as displayed in Figure 3. To achieve this, we preform spherical interpolation between both the initial Gaussian noises and the Langevin noises at each sampling step.

## 6 CONCLUSION

We demonstrate a novel paradigm, inspired by energy-based models and diffusion-based models, that aims to implicitly learn intermediate latent distributions without explicitly defining a noise degradation schedule. This removes the inaccuracies of approximating transitional distributions with Gaussian noise, allowing for a shorter and more efficient sampling. With the help of gradient penalty regularization, our energy model is capable of learning a sequence of energy functions that better guide the Langevin dynamics sampling process. Through our experiments, we show that our model is capable of generating high quality images on diverse datasets. In future work it is desired to explore new methods for more accurate sampling from energy models, which could help methods such as ours scale better for higher numbers of sampling steps.

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

# A APPENDIX

## A.1 TRAINING HYPERPARAMETER DETAILS

In Table 7 we present detailed hyperparameter specifications for the neural network architectures we use. For all of our evaluations we trained the models on 4 Nvidia GeForce RTX 3090 GPUs. We initially tuned $\eta$ so that the energy of generated samples is slightly bellow the energy of real images at the final sampling step. In Figure 4 we plot a comparison of energy predictions when $\eta$ is slightly larger or smaller than the value we suggest. Smaller values typically lead to noisy results, as the Langevin process hasn't fully converged, while higher values yield sampling artifacts and high contrast.

## A.2 MORE QUALITATIVE EXPERIMENTS

Additional uncurated samples generated by our best CIFAR-10 model are provided in Figure 5. We also conduct a similarity comparison to demonstrate that our method can generalize well. We use cosine similarity to determine the nearest neighbours for our generated samples. The results can be viewed in Figures 6 to 9, done on the CelebA Liu et al. (2015), LSUN-Churches Yu et al. (2015), AFHQV2 Choi et al. (2020) and LSUN-Conference Rooms datasets respectively.

Table 7: Hyperparameters for training our energy-based model.

|  | Resnet-Based (32x32) | Unet-Based (32x32) | Unet-Based (64x64) |
|---|---|---|---|
| Sampling steps ($T$) | 20 | 20 | 10 |
| Langevin dynamics steps ($K$) | 3 | 3 | 6 |
| Step size ($\eta$) | 1.7 | 1.7 | 2.89 |
| Starting noise ($\sigma$) | 0.3 | 0.3 | 0.3 |
| Batch size | 256 | 256 | 128 |
| Size of fake buffer ($\mathcal{B}$) | 10k | 10k | 10k |
| Weight decay | - | 0.01 | 0.01 |
| Channels | 128 | 128 | 128 |
| Channel multipliers | 1,1,1,2,2,2 | 1,2,2,1 | 1,2,3,4 |
| Heads | - | 2 | 2 |
| Blocks per resolution | - | 2 | 2 |
| Attention at resolutions | - | - | - |
| Model Parameters | 6M | 9M | 45M |

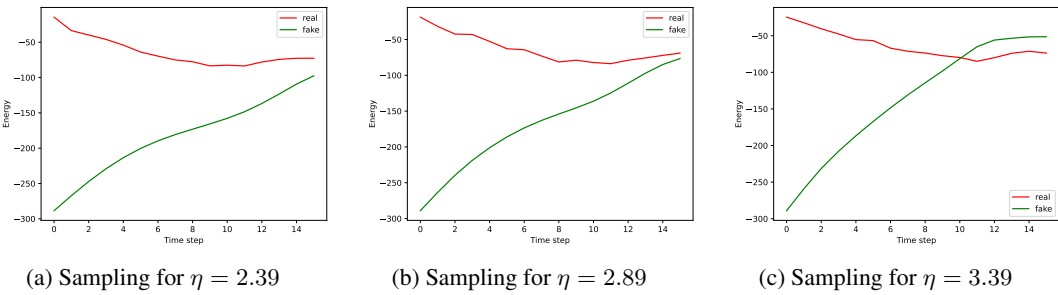

(a) Sampling for $\eta = 2.39$  (b) Sampling for $\eta = 2.89$  (c) Sampling for $\eta = 3.39$

Figure 4: Comparison of energy values predicted by our model for slightly different values of $\tilde{\alpha}$ when trained on CelebAHQ $64^2$

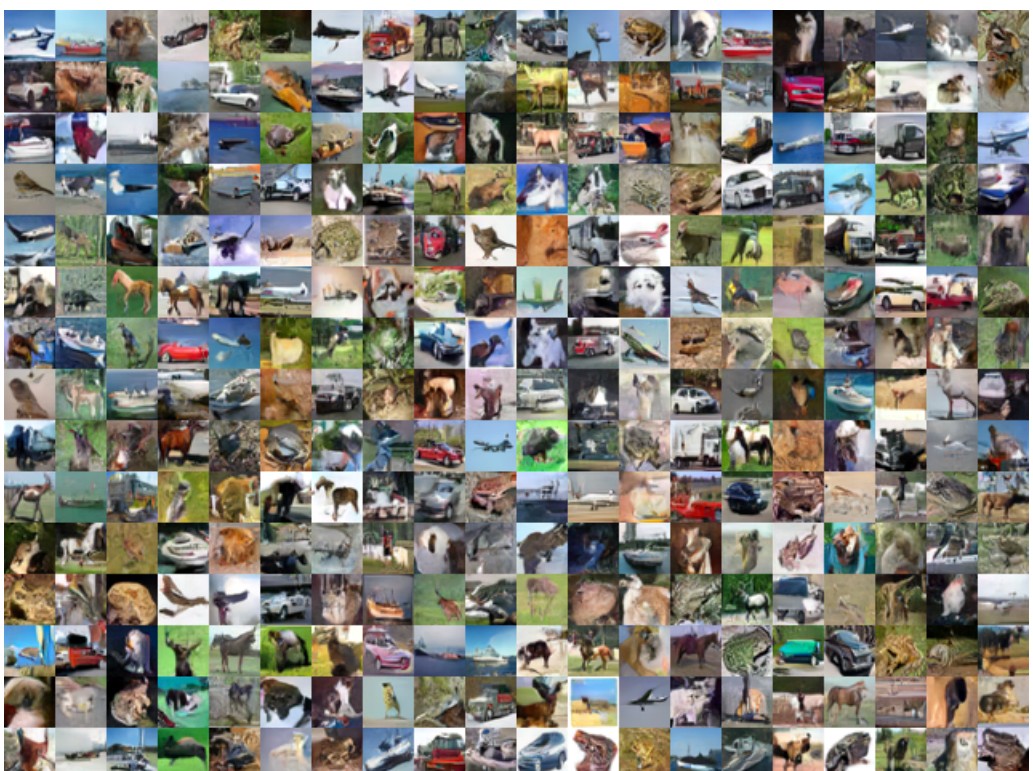

Figure 5: Uncurated generated samples on CIFAR-10.

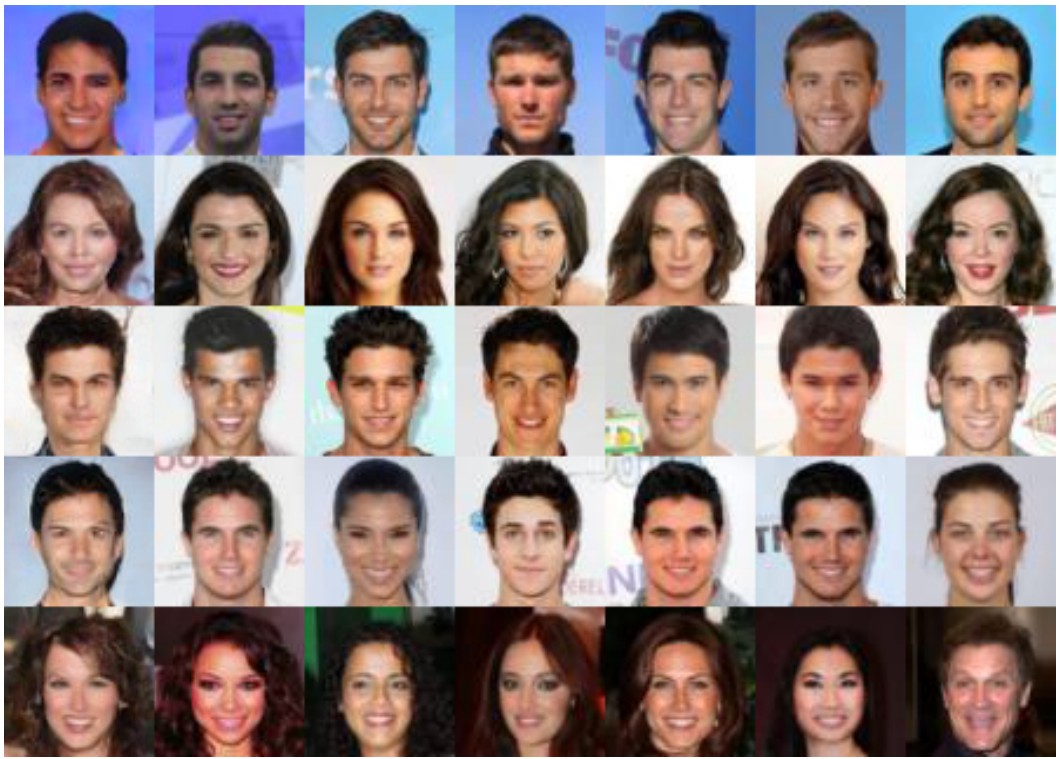

Figure 6: Cosine similarity comparison on generated images (leftmost column) with the closest real images from CelebA

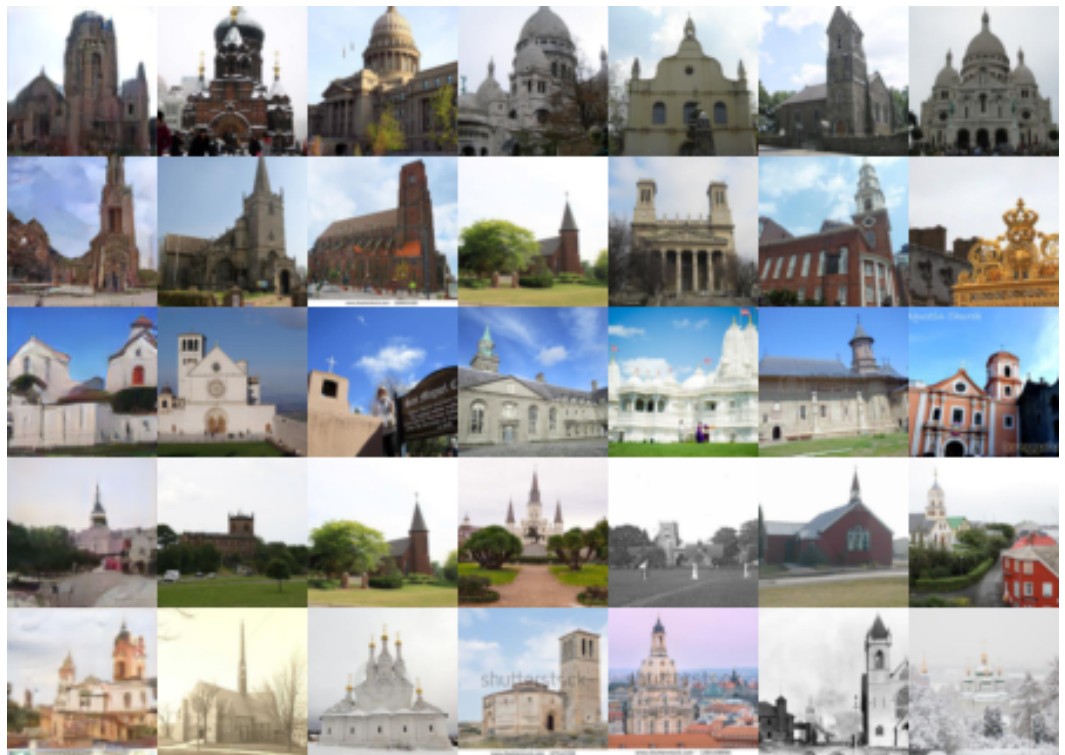

Figure 7: Cosine similarity comparison on generated images (leftmost column) with the closest real images from LSUN churches

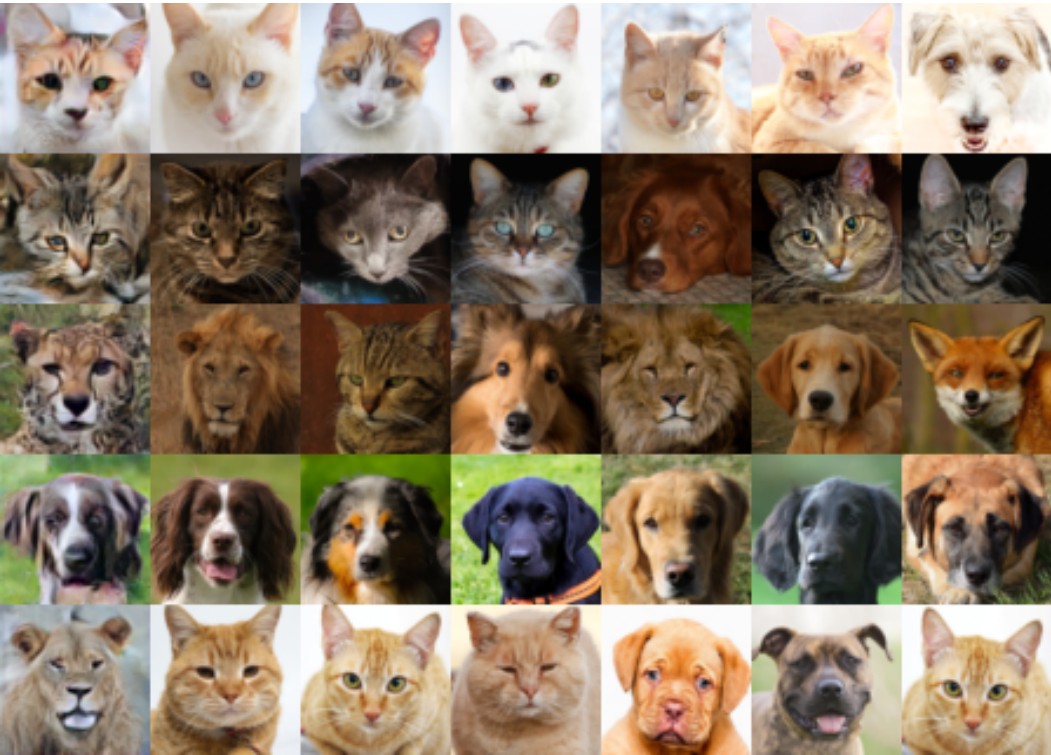

Figure 8: Cosine similarity comparison on generated images (leftmost column) with the closest real images from AFHQV2

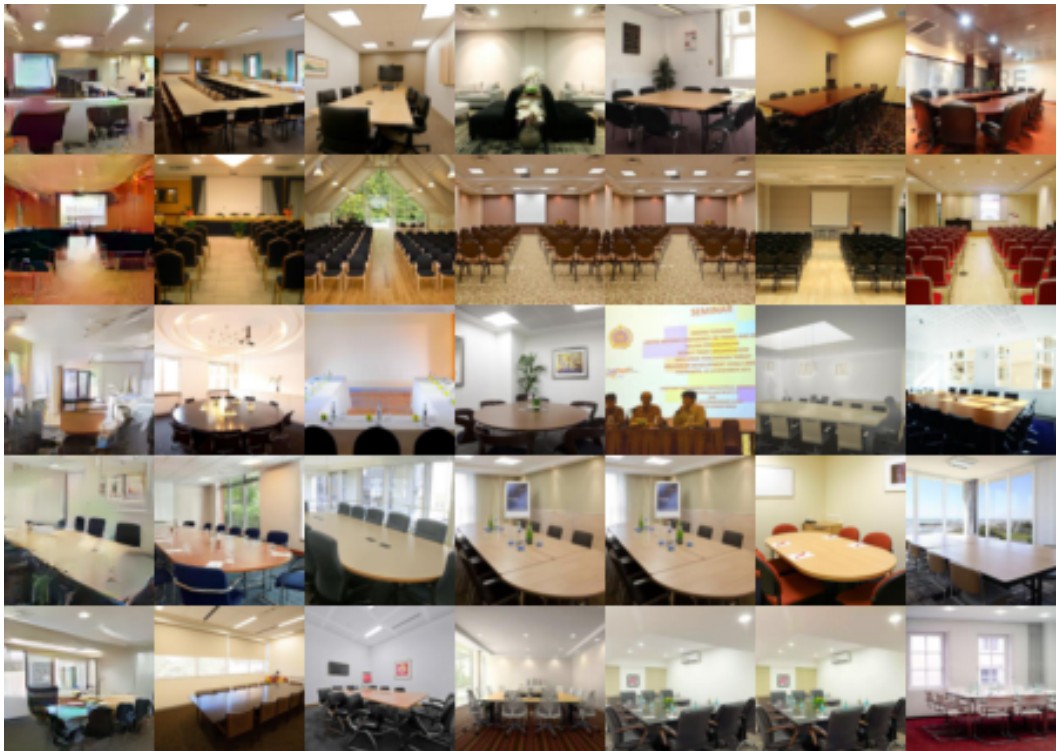

Figure 9: Cosine similarity comparison on generated images (leftmost column) with the closest real images from LSUN conference rooms

