# OpenReview forum: "Diffusion Process with Implicit Latents via Energy Models"
_ICLR.cc/2025/Conference — ICLR 2025 Conference Withdrawn Submission_

### Official Review · Reviewer_Vu3o · 2024-10-29

**Soundness:** 3
**Presentation:** 2
**Contribution:** 2
**Rating:** 5
**Confidence:** 3

**Summary:**

This paper proposes a new method for training energy models based on Langevin dynamics. Instead of a predefined fixed noise schedule, the intermediate transitions are implicitly defined.

**Strengths:**

This new method removes the need for a predefined diffusion noise schedule, offering the possibility of finding better intermediate latents. Evaluation compares to other methods.

**Weaknesses:**

Writing could be clearer (see questions below) (improved clarity might address some of my other concerns). Seems to combine fairly standard concepts like energy models, contrastive loss, and Langevin dynamics into a new proposed algorithm but the advantages in performance are not entirely clear to me (apparently faster sampling than baseline EBMs with similar quality, but quality still falls very short of diffusion models).

**Questions:**

First, it is not clear to me how the inference procedure works for this model. Do you run 1 (or more) steps of LD at each iteration, but using the time-dependent learned score f_\theta(x_t, t)? Would be nice to have the inference algorithm clearly stated, like the Training algorithm (Alg 1). Also, what is the intuition for why inference should require fewer steps? (Is it because you remove the assumption of Gaussian transitions?)

Second, can you clarify the advantages of energy parametrization vs score parametrization for your setting? It seems to me that the idea of implicit latents does not depend on EBM and could also work with score? EBMs are interesting and worth exploring but typically their performance lags behind score parametrization, as reflected in your experiments. For example in Table 1, your method achieves FID around 17, which is not bad for an EBM but much worse the the FID ~3 achieved by DDPM. I wonder what makes EBMs fundamentally better for the problem you're trying to solve?

Section 4.1: I’m confused about the motivation of the algorithm, and in particular whether t represents a time or an index into the Langevin step (or both?). In eq. (9) it looks like x_{t-1} is obtained by running one step of LD on x_t using the f_\theta(x_t, t). If f_\theta did not depend on time this would be a standard LD iteration, and the 'intermediate latents' would simply the distributions you would obtain at each step of LD, starting from Gaussian noise. However f does depend on t, and it seems that you are also treating t as representing something analogous to the time in diffusion. It is unclear to me exactly how/why this helps.

In eq. (10) it looks like the target distribution for x_t is always just the data distribution? How do we get ‘useful’ latents (i.e. that do some
kind of annealing) in that case? Is it because of the limited ability of f(x_t, t) to actually predict x_0 given x_t? (similar to how “x0-prediction” in diffusion actually predicts E[x_0 | x_t])? Would appreciate clarification.

Minor things:
L148-9: shouldn't energy should be *low* when x belongs to the distribution?

Section 4.2: This is a fairly standard calculation in diffusion e.g. Luo https://arxiv.org/abs/2208.11970 (unless there is a difference I’m not seeing?) so might be better in an appendix — leaving more room to clarify other things.

L267: I think you mean second term.

---

### Official Review · Reviewer_unGR · 2024-11-05

**Soundness:** 2
**Presentation:** 2
**Contribution:** 1
**Rating:** 3
**Confidence:** 3

**Summary:**

The paper proposes a generative model combining diffusion-based processes and energy-based models (EBMs) to implicitly learn intermediate latent distributions without a predefined noise degradation schedule. It claims improvements in image generation quality and sampling efficiency across datasets like CIFAR-10, CelebA, and LSUN. However, while this paper introduces an intriguing combination of techniques, the practical results are underwhelming. Performance is only marginally improved, if at all, over comparable models, and the method’s complexity may not be justified by its empirical gains. The paper would benefit from additional baseline comparisons, improvements in performance, and a clearer justification of its technical choices.

**Strengths:**

**1. Innovative Approach**: The proposed integration of EBMs with diffusion-based techniques to learn latent distributions adaptively shows originality, particularly in attempting to sidestep traditional noise scheduling.

**2. Technical Basis**: The use of Langevin dynamics and WGAN-LP as stabilizing techniques is appropriate and aligns with best practices in EBM training. The exploration of adaptive, implicit intermediate distributions is theoretically interesting.

**Weaknesses:**

**1. Underwhelming Performance**: Despite the innovation, the model’s empirical performance is not close to state-of-the-art results. The fidelity improvements on datasets like CIFAR-10 and CelebA are marginal compared to existing models optimized for both quality and efficiency (e.g., recent diffusion and GAN-based approaches). This raises questions about the model’s practical value.

**2. Limited Improvement in Sampling Efficiency**: While the model aims to reduce inference steps, the actual reduction is modest. Competing approaches, particularly those using optimized sampling schedules (e.g., DDIM), achieve comparable or better efficiency with similar or higher image quality.

**3. Methodological Clarity and Justification**: Some technical choices, such as the decision to omit a generator and rely solely on energy functions for latent representations, seem theoretically sound but do not lead to significant performance gains. A clearer justification is needed for how this choice benefits practical application, given the weak empirical results.

**4. Insufficient Baselines and Metrics**: The paper does not adequately compare with recent efficient sampling techniques (such as DDIM), and relies on FID and Inception scores. Incorporating further comparisons and more diverse evaluation metrics (e.g., perceptual similarity measures) would strengthen the empirical analysis.

**Questions:**

please see the weaknesses.

---

### Official Review · Reviewer_VwYc · 2024-11-06

**Soundness:** 2
**Presentation:** 2
**Contribution:** 1
**Rating:** 3
**Confidence:** 4

**Summary:**

# Summary
This paper introduces a method for training Energy-Based Models (EBMs) without needing to specify a generator for intermediate latent states encountered during sampling. The proposed algorithm addresses this by unrolling the sampling process from noise to clean input, generating the intermediate samples. This approach also contrasts with denoising diffusion probabilistic models (DDPMs), where intermediate states are explicitly defined through a sequential noise degradation process.

# Strengths
Typically, Energy-Based Models require a generator since intermediate distributions are not directly defined in the training phase. However, the proposed algorithm addresses this by unrolling the sampling process from noise to clean input, imposing a reconstruction loss between the generated input and the ground truth.


# Weaknesses

## Concern 1: Weak Empirical Results
(1) The trained model does not achieve faster sampling or superior sample quality. For instance, older methods such as Karras et al. (2020) and CTM (Ho et al., 2020) significantly outperform this approach, achieving FIDs of approximately 3 compared to around 17 for the proposed method on CIFAR-10. Additionally, more recent models like CTM (Dongjun et al., 2023) reach FID scores below 2 with one-step generation on CIFAR-10. Consequently, this method appears to lack practical significance.

(2) This algorithm is likely to incur longer training times since it requires backpropagation through time as the sample generation process is unrolled from noise to clean input. How do the model size and training iterations compare with those of diffusion and EBM baselines, specifically those in Gao et al. (2020a, 2022b)?

## Concern 2: Weak Theoretical Contribution
While achieving top performance metrics is not the sole criterion for acceptance, this paper also lacks a substantial theoretical contribution. For example, the connection to diffusion models feels tenuous, as no corruption occurs during the forward process, which doesn’t impact the sampler. Additionally, the learning objective has already been proposed in prior work (Xie et al., 2016b) without a connection to diffusion models.

Typos:
Eqn (5) should be $exp(f_\theta(x))$

---
# Citations

[1] Ruiqi Gao, Yang Song, Ben Poole, Ying Nian Wu, and Diederik P Kingma. Learning energy-based models by diffusion recovery likelihood. arXiv preprint arXiv:2012.08125, 2020b.

[2] Ruiqi Gao, Erik Nijkamp, Diederik P Kingma, Zhen Xu, Andrew M Dai, and Ying Nian Wu. Flow contrastive estimation of energy-based models. 2020 ieee. In CVF Conference on Computer Vision and Pattern Recognition (CVPR), pp. 7515–7525, 2020a.

[3] Jianwen Xie, Yang Lu, Song-Chun Zhu, and Yingnian Wu. A theory of generative convnet. In International Conference on Machine Learning, pp. 2635–2644, 2016b.

[4] Kim, Dongjun, Chieh-Hsin Lai, Wei-Hsiang Liao, Naoki Murata, Yuhta Takida, Toshimitsu Uesaka, Yutong He, Yuki Mitsufuji, and Stefano Ermon. "Consistency trajectory models: Learning probability flow ode trajectory of diffusion." arXiv preprint arXiv:2310.02279 (2023).

**Strengths:**

see summary

**Weaknesses:**

see summary

**Questions:**

see summary

---

### Note · Authors · 2024-11-14

I have read and agree with the venue's withdrawal policy on behalf of myself and my co-authors.